# Optimized Protocol for Isolation of Small Extracellular Vesicles from Human and Murine Lymphoid Tissues

**DOI:** 10.3390/ijms21155586

**Published:** 2020-08-04

**Authors:** Marie Bordas, Géraldine Genard, Sibylle Ohl, Michelle Nessling, Karsten Richter, Tobias Roider, Sascha Dietrich, Kendra K. Maaß, Martina Seiffert

**Affiliations:** 1Division of Molecular Genetics, German Cancer Research Center (DKFZ), 69120 Heidelberg, Germany; m.bordas@dkfz.de (M.B.); s.ohl@dkfz.de (S.O.); 2Faculty of Biosciences, University of Heidelberg, 69120 Heidelberg, Germany; 3Division of Biomedical Physics in Radiation Oncology, German Cancer Research Center (DKFZ), 69120 Heidelberg, Germany; g.genard@dkfz-heidelberg.de; 4Central Unit Electron Microscopy, DKFZ, 69120 Heidelberg, Germany; m.nessling@dkfz.de (M.N.); k.richter@dkfz.de (K.R.); 5Department of Medicine V, Hematology, Oncology and Rheumatology, University of Heidelberg, 69120 Heidelberg, Germany; Tobias.Roider@med.uni-heidelberg.de (T.R.); Sascha.Dietrich@med.uni-heidelberg.de (S.D.); 6Hopp-Children’s Cancer Center Heidelberg (KiTZ), 69120 Heidelberg, Germany; k.maass@kitz-heidelberg.de; 7Division of Pediatric Neurooncology, German Cancer Research Center (DKFZ), 69120 Heidelberg, Germany

**Keywords:** extracellular vesicles, exosomes, small extracellular vesicles, isolation, purification, size-exclusion chromatography, ultracentrifugation, sucrose density cushion, lymph node, spleen, solid tissue

## Abstract

Small extracellular vesicles (sEVs) are nanoparticles responsible for cell-to-cell communication released by healthy and cancer cells. Different roles have been described for sEVs in physiological and pathological contexts, including acceleration of tissue regeneration, modulation of tumor microenvironment, or premetastatic niche formation, and they are discussed as promising biomarkers for diagnosis and prognosis in body fluids. Although efforts have been made to standardize techniques for isolation and characterization of sEVs, current protocols often result in co-isolation of soluble protein or lipid complexes and of other extracellular vesicles. The risk of contaminated preparations is particularly high when isolating sEVs from tissues. As a consequence, the interpretation of data aiming at understanding the functional role of sEVs remains challenging and inconsistent. Here, we report an optimized protocol for isolation of sEVs from human and murine lymphoid tissues. sEVs from freshly resected human lymph nodes and murine spleens were isolated comparing two different approaches—(1) ultracentrifugation on a sucrose density cushion and (2) combined ultracentrifugation with size-exclusion chromatography. The purity of sEV preparations was analyzed using state-of-the-art techniques, including immunoblots, nanoparticle tracking analysis, and electron microscopy. Our results clearly demonstrate the superiority of size-exclusion chromatography, which resulted in a higher yield and purity of sEVs, and we show that their functionality alters significantly between the two isolation protocols.

## 1. Introduction

Extracellular vesicles (EVs) are lipid bilayer-enveloped nanovesicles secreted by both eukaryotic and prokaryotic cells and carrying cargos of proteins, lipids, and nucleic acids [1,2]. EVs contain both surface and luminal factors which can be used as markers for specific EV populations representing the different biogenesis pathways [3,4]. Although the definition of EVs is continuously being refined, currently three main subtypes of eukaryotic cell-derived EVs can be distinguished based on their size, composition, and cellular origin—small EVs (sEVs or exosomes, 30–150 nm), microvesicles (MVs, 100 nm^−1^ µm), and apoptotic bodies (1–5 µm) [5,6]. Unlike MVs, which originate from direct budding of the plasma membrane, sEVs stem from the endocytic compartment and are released after fusion of multivesicular bodies with the plasma membrane [5,7]. Due to the secretory release mechanism of MVs, it is well recognized that their cargo mirrors the cytoplasmic and surface composition of the parental cell. In contrast, several studies on sEV loading reported a specific enrichment or depletion of cellular proteins or RNAs in their cargoes, and several sorting mechanisms have been suggested [8,9,10,11,12].

sEVs have been shown to be taken up by various recipient cell types such as myeloid, stromal, and neuronal cells, among many others [13]. The delivery of sEV cargoes into recipient cells can lead to both transcriptional and proteomic changes as a result [1,14,15]. Depending on the origin of the sEVs and the recipient cells, sEV uptake can affect diverse biologic processes, e.g., inflammation, angiogenesis, immune response, or composition of the extracellular matrix [1]. Besides their functional properties, sEVs and their content, in particular microRNAs, are also discussed for their potential as diagnostic and prognostic biomarkers in pathological conditions [1,16]. More recently, researchers explored sEVs as a new therapeutic tool for targeted drug delivery [17,18].

Due to their large spectrum of action, the interest of the scientific community for sEVs has increased exponentially over the last few years. However, many technical limitations are encountered during isolation and purification of sEVs. In particular, the isolation of sEVs from solid tissues remains challenging, limiting studies with primary patient material and causing a biased use of cell line-derived sEVs. To overcome this limitation, we aimed to improve the isolation and purification of sEVs from lymphoid tissues of human and murine specimens by comparing two different isolation protocols. The first protocol is based on differential centrifugation combined with ultracentrifugation on a sucrose density cushion as previously described [19], whereas the second protocol combines differential centrifugation with size-exclusion chromatography (SEC) using the commercially available single qEV 35 nm columns from IZON (Izon, Christchurch, New Zealand) [20]. Previous studies have already compared the efficiency of qEV IZON columns with other accepted sEV isolation techniques and reported higher yields and quality of the final product, in particular for isolation of sEVs from plasma samples [21,22].

As starting material, we used three biopsies of lymph nodes (LNs) collected from patients with B-cell lymphoma and three spleens from a B-cell lymphoma mouse model [23,24]. By directly comparing the amount, purity, and functionality of sEVs obtained for both sample types with the two protocols, we demonstrate the superiority of the SEC-based isolation technique for lymphoid tissues.

## 2. Results

### 2.1. Isolation and Purification of sEVs from Human Lymph Nodes

Two protocols for sEV isolation from lymphoid tissues were performed in parallel on the same starting material to compare their efficiency in terms of (1) total amount of recovered sEVs, (2) purity of sEV preparation, and (3) reproducibility. After manual dissociation of LN biopsies of three B-cell lymphoma patients, the supernatants of the cell suspensions were collected and processed by differential centrifugation. The resulting pellets (100 K pellet) containing sEVs and soluble proteins and lipids was resuspended and split into two equal parts each, which were then combined either with SEC on IZON columns or differential centrifugation combined with ultracentrifugation on a 40% sucrose density cushion as illustrated in Figure 1. An identical volume of PBS (250 µL) was used for the final resuspension of sEVs isolated from IZON columns and the sucrose density cushions (“cushion”). Nanoparticle tracking analysis (NTA) revealed that the resuspended pellet from the “cushion” preparation as well as fractions 1 and 2 collected from the IZON column were enriched in the characteristic sEV size profile, with IZON fraction 2 accounting for the peak fraction (Figure 2A, left). SEV size profiles were also detected in the IZON fraction 3, although in lower concentrations. The absolute number of particles yielded from IZON peak fractions as assessed by NTA was 3.9- to 10.3-fold higher than the sEV particle number recovered using the sucrose density cushion (Figure 2A, right). We then performed protein quantification using a bicinchoninic acid (BCA) assay (Figure 2B, left). Due to their smaller size, protein complexes are able to enter the pores of the IZON column, and their elution is delayed, which can be observed as a second protein peak in the fraction F7 collected later [20,25]. The absolute amount of proteins recovered from IZON peak fractions was lower than the one obtained in the respective “cushion” preparation (1.4- to 2.2-fold lower; Figure 2B, right) which was less than the fold change detected by NTA for particle numbers. Calculation of the particle/protein ratios revealed lower values for “cushion” preparations compared to IZON fractions in two of the three samples (Figure 2D). Therefore, we hypothesized that the sucrose density cushion approach led to a larger amount of protein complexes co-isolated with the sEVs. The mean particle size and size distribution of sEVs were similar in IZON fractions 1 and 2 and the “cushion” preparation with 153, 157, and 148 nm for the peak IZON fractions, and 155, 163, and 152 nm for the corresponding “cushion” preparations (Figure 2C,E). In line with our hypothesis, immunoblot analysis revealed a lower signal for exosomal surface markers FLOTILLIN-1, CD81, CD9, and the luminal marker TSG101 in the “cushion” preparations compared to IZON fractions 1 and 2 for the same amount of protein loaded (Figure 2F and Appendix A). As our study is one of the first to focus on solid tissues, we thoroughly validated the presence of contaminant proteins as recommended by the MISEV guidelines [26]. We neither detected the Golgi marker GM130 nor the mitochondrial marker CYTOCHROME C in both preparations. Surprisingly, we detected the endoplasmic reticulum (ER) protein CALNEXIN in sEVs isolated with both protocols. However, the amount of CALNEXIN in the sEV preparations was lower in comparison to the parental cell lysate (Figure 2F and Appendix A). Although partial contamination of the samples with cellular debris cannot be excluded, the presence of CALNEXIN but no markers from other cell organelles might be indicative for a specific sEV biogenesis pathway involving the ER in lymphoma cells. In addition, the IZON fraction F2 and “cushion” samples were analyzed by transmission electron microscopy (TEM). The results illustrate that the SEC isolated samples allow a clear identification of sEVs for all of the three samples. However, we observed a higher heterogeneity in the “cushion” preparations, with sEVs barely detected in two out of three samples (Figure 2G). Employing immuno-electron microscopy, we confirmed the presence of the immune receptor MHC Class II (HLA-DR) on the surface of sEVs isolated by both approaches and thereby validate their immune cell origin (Figure 2H).

### 2.2. Isolation and Purification of sEVs from Murine Spleen

Spleens from three mice with B-cell lymphoma were dissociated and processed as outlined in Figure 1. Similar to human LN samples, NTA results revealed an enriched particle concentration in IZON fractions 1 and 2, with fraction 2 being the peak fraction (Figure 3A, left). For one of the samples, sEVs were mainly detected in fractions 2 and 3, a difference we attribute to manual loading and elution of the IZON column. The absolute number of particles isolated was 4.8- to 27.7-fold higher in the IZON peak fractions in comparison to the respective cushion preparations (Figure 3A, right). The protein concentrations measured by BCA were more similar between IZON fractions and “cushion” and might be attributed to protein complexes co-isolated with the sEVs in the “cushion” preparation (Figure 3B, left). The absolute amount of proteins recovered was 1.4- to 2.6-fold higher in the IZON preparations in comparison to the respective cushion preparations (Figure 3B, right). In line with these results, the particle/protein ratios were drastically reduced for “cushion” preparations in comparison to IZON fractions (Figure 3D). The mean particle sizes were 154, 142, and 157 nm in the IZON peak fractions, and 114, 143, and 155 nm for the corresponding “cushion” preparations (Figure 3C). Those results imply that, for one preparation at least, the obtained product was different when using the SEC or the sucrose density cushion approach. Additionally, we observed a difference in the size distribution profile depending on the isolation protocol used, which might be explained by different EV subpopulations isolated by the different approaches (Figure 3E). Immunoblot results confirmed the exosomal identity of the particles in fractions 1, 2, and 3 and the cushion fraction by positive bands for the surface marker FLOTILLIN-1 but also the luminal markers ALIX and TSG101 in the IZON fractions 1–3 and in the “cushion” preparations (Figure 3F and Appendix A). FLOTILLIN-1 was only detected in IZON fractions, and TSG101 showed varying intensities being highly present in the “cushion” fraction while only weakly detected in the IZON fractions. Together with the variance in NTA size profiles, the immunoblotting results further suggested that different sEV subpopulations were isolated. Irrespective, we could exclude mitochondrial contaminations by ATP5A being absent from all sEV fractions (Figure 3F and Appendix A). In concordance with the human LN data, CALNEXIN could again be detected in sEV fractions from both protocols. The quality and purity of the sEV isolations was further assessed by TEM (Figure 3G). We noted a high heterogeneity among the samples for the “cushion” preparations, with one sample highly enriched in lipidic structures (Figure 3G). Interestingly, we noticed the recurrent presence of small dark structures of an approximate size of 10 nm exclusively in “cushion” preparations (Black arrows, Figure 3G). A closer look at the particles revealed a specific geometrical shape, typical for ferritin (Figure 3H) [27,28]. We further observed a red color of the sEV pellets and suspensions which is typical for a contamination with erythrocyte-derived protein, strengthening our hypothesis (Figure 3I).

### 2.3. Functional Analysis of sEVs Isolated by the Two Different Protocols

We and others have previously reported that tumor-derived sEVs (TEX) are able to induce an immunosuppressive phenotype in monocytes in vitro, with a typical upregulation of surface PD-L1 and HLA-DR [29,30,31]. We compared the potential of murine TEX isolated from three spleen samples using the two different approaches regarding their ability to induce such a phenotype. Both TEX preparations (IZON and “cushion”) induced PD-L1 upregulation in monocytes, although to various degrees (Figure 4A and Appendix A for gating strategy). However, for two of the three samples, “cushion” preparations did not induce an upregulation of HLA-DR (Figure 4B). These results indicate that both protocols resulted in sEV samples that induce a different immunosuppressive phenotype in monocytes. We also analyzed the expression of the activation marker ICAM-1 (CD54) in monocytes treated with TEX, which showed a much more drastic upregulation with the “cushion” preparations compared to the SEC-isolated TEX in all 3 analyzed samples (Figure 4C).

## 3. Discussion

Multiple isolation approaches have been proposed for sEV preparations, including commercially available kits, ultrafiltration, polymer precipitation, immune-affinity capture, size-exclusion chromatography, ultracentrifugation, and ultracentrifugation combined with density cushion [32,33]. The selection of the isolation technique must consider the subsequent usage of the sEV preparations. Yield is generally prioritized when performing RNA or DNA sequencing. However, contamination with protein or lipid complexes must be avoided for proteomic analysis or functional assays. The sources and risk of protein contamination are even higher when isolation of sEVs is performed from solid tissues that require mechanical or enzymatic dissociation. Here, we compared two different protocols to isolate sEVs from solid lymphoid tissues: differential centrifugation combined to SEC using commercially available IZON columns and differential centrifugation combined to ultracentrifugation on sucrose density cushions. Although a total of three human LN and three mouse spleens are shown in our manuscript, our results are representative of larger cohorts of samples regularly analyzed in our laboratory. Both approaches led to efficient isolation of sEVs as shown by size characterization based on NTA analysis and the presence of exosomal markers by immunoblotting. However, further characterization of the preparations using BCA assay, TEM, and functional assay led us to conclude that the SEC approach is superior in terms of purity, quantity, and reproducibility.

In particular, our results strongly suggest that isolation of sEVs by the sucrose density cushion isolation protocol results in a more severe co-isolation of protein complexes with the sEVs. Using immunoblotting, we excluded contamination by mitochondrial and Golgi-derived proteins. We speculate that the presence of cellular debris in the supernatant of the dissociated tissues, which would lead to the sample contamination, was efficiently avoided by the rapid isolation of sEVs following organs’ resection. However, the ER-derived protein CALNEXIN was found in sEV preparations using both isolation approaches. Such contamination likely results from the tissue dissociation. However, further investigations are required to verify that the presence of ER-proteins but not of proteins of other organelles could be the result of a specific packaging mechanism of tumor-derived sEVs. Furthermore, we also observed the presence of ferritin-like proteins in spleen sEVs isolated by the sucrose density cushion but not the SEC approach. The presence of ferritin seems to indicate an erythrocyte contamination. However, addition of erythrocyte lysis buffer to the supernatant would result in an increased release of hemoglobin. As erythrocytes are easier to separate from sEVs than hemoglobin, we do not recommend the usage of such buffer.

Previous studies focusing on plasma-derived sEVs reported lipoproteins as the main contaminants of sEV preparations [34,35,36,37]. Unfortunately, lipoproteins cannot be efficiently discriminated from sEVs when performing NTA analysis. However, contamination by lipoproteins of low and high density in sEV preparations seem less important when using the SEC-isolation approach [34,35,36]. In our study, we noticed the presence of large lipidic structures in one of the three murine samples isolated with the sucrose density cushion but not with SEC. Additional immunoblots are required to conclude on lipoprotein contamination in sEVs isolated with both approaches. A solution to limit lipoprotein contamination would be the combination of both SEC and sucrose density cushion. However, combination of isolation methods often results in a drastic loss of material. Other possible sources of contamination include secreted proteins, and extracellular matrix proteins. Investigations on such contaminants remain challenging, as these proteins could be considered as well as of exosomal origin.

We also would like to emphasize that ultracentrifugation combined with density cushion and differential centrifugation combined with SEC are isolation techniques that are based on the density or the size of EVs, respectively. Thus, it is possible that the use of a unique isolation protocol may impact on the distribution of sEV subpopulations in the preparations. In line with this hypothesis, different sEV marker proteins were enriched in sEVs isolated from spleens by the two different methods: sEV preparations obtained using the SEC approach were enriched in FLOTILIN-1 and ALIX but not TSG101, whereas the “cushion” preparations did show an enrichment in ALIX and TSG101 but not in FLOTILIN-1.

We previously reported that treatment of monocytes with TEX induces the upregulation of surface PD-L1 and major histocompatibility complex (MHC) II/HLA-DR. We compared the capability of TEX preparations of both isolation protocols to induce such a phenotype, using an identical amount of sEVs based on protein quantification. Treatment of monocytes with “cushion” preparations resulted in a more heterogeneous response of those two markers in comparison to SEC preparations, indicating different amounts of contaminant proteins from one “cushion” preparation to another. In particular, MHC II surface expression was increased when monocytes were treated with IZON preparations but not with “cushion” preparations, for two of three preparations. Yet, it is known that sEVs secreted by antigen-presenting cells are enriched in MHC II molecules, and that sEVs can promote the transfer of functional MHC II/antigen complexes to recipient cells [38,39]. On the contrary, we observed a higher upregulation of the monocyte activation marker ICAM-1 upon treatment with “cushion” preparations. These results highlight that contaminant proteins can interfere with biological results and lead to an incorrect conclusion of sEV-induced phenotypes. ICAM-1 expression on monocytes is a general activation marker and its upregulation can be induced by cytokines, lipoproteins, LPS etc. [40,41]. Given these results, we suspect that the PD-L1 upregulation observed in monocytes treated either with the IZON preparations or the “cushion” preparations is the consequence of monocytes’ activation mainly by sEVs, whereas induced expression of ICAM-1 results from both sEVs and non-sEV contaminants. These results raise the hypothesis that soluble pro-inflammatory cytokines, secreted in B-cell lymphoma microenvironments, might contribute to the contamination in the “cushion” preparations, although further investigations would be required for firm conclusion.

As a conclusion, we strongly recommend the usage of SEC for sEV isolation from solid tissue represented here by lymphoid tissues. Multiple controls should be performed to validate the purity of the samples. Such controls include extensive immunoblotting of positive and negative exosomal markers. Reaching a complete purity of sEVs from biofluids or solid tissue seems unrealistic. Nevertheless, immunoblotting results in parallel to NTA analysis can provide a reliable estimation of preparations’ contamination by protein complexes. TEM remains an indispensable tool to validate the presence and integrity of sEVs and to assess the amount of contamination by lipid complexes.

## 4. Materials and Methods

### 4.1. Animals

Eμ-TCL1 mice on C57BL/6 background were kindly provided by Carlo Croce (Ohio State University). C57BL/6 wild-type (WT) mice were purchased from Charles River Laboratories (Sulzfeld, Germany). Adoptive transfer of Eµ-TCL1 tumors was performed as previously described [42,43]. Briefly, 1–2 × 10^7^ B-cells enriched from Eμ-TCL1 splenocytes were transplanted intraperitoneally (i.p.) into C57BL/6N WT animals. B-cell enrichment was performed using EasySep™ Mouse Pan-B Cell Isolation Kit (Stemcell Technologies, Vancouver, BC, Canada), yielding a purity above 95% of CD5+ CD19+ cells. Tumor load was assessed in the blood every week using flow cytometry as the proportion of CD5+ CD19+ cells among CD45+ cells. Animals with a tumor load >90% in peripheral blood were sacrificed; spleen was isolated and mechanically dissociated in PBS. All animal experiments were carried out according to institutional and governmental guidelines approved by the local authorities (Regierungspräsidium Karlsruhe, permit number G98/16, approved on 13 July 2016).

### 4.2. SEV-Free RPMI Medium

Fetal calf serum (FCS) (Gibco, Carlsbad, CA, USA) was ultra-centrifuged at 100,000× *g* for 18 h at 4 °C. FCS supernatant was filtered through a 0.22 µm filter. RPMI 1640 medium (Thermo Fisher Scientific Inc., Waltham, MA, USA) was supplemented with 10% sEV-free FCS and 1% penicillin/streptomycin (Gibco, Carlsbad, CA, USA). The medium was filtered through a 0.22 µm filter prior to use.

### 4.3. Isolation of Lymph Node Supernatants

Patient lymph node (LN) samples were obtained after the study protocols’ approval by local ethics’ committees from the Department of Medicine V of the University Clinic Heidelberg according to the declaration of Helsinki, and with patients’ informed consent. LN samples were collected directly after biopsies from patients with diverse B-cell lymphomas. LNs were placed in 0.9% NaCl solution and processed immediately. Each LN was cut in small pieces with a maximum size of 2 mm. Cells were released in 50 mL of RPMI medium supplemented with sEV-free FCS (10%), Penicillin-Streptomycin (1%) and l-Glutamine (1%).

### 4.4. Isolation of Murine Spleen Supernatants

Entire spleens from three adoptively transferred Eμ-TCL1 mice were collected in 7 mL of 0.22 µm-filtered PBS each. Spleens were mechanically dissociated using MACS dissociator (Miltenyi Biotec, Bergisch Gladbach, Germany), using the program “m_spleen_01”.

### 4.5. Differential Centrifugation

Collected supernatants of human LN and mouse spleen were centrifuged at 300× *g* for 10 min at 4 °C in a swing-out centrifuge to remove cellular debris. Resulting supernatants were transferred into new collection tubes and centrifuged at 2000× *g* for 20 min at 4 °C to remove larger apoptotic bodies. Resulting 2000× *g* supernatants were transferred into new collection tubes and centrifuged at 10,000× *g* for 40 min at 4 °C to remove MVs. Resulting 10,000× *g* supernatants were transferred into ultracentrifugation tubes (#5031, Seton Sci., Petaluma, CA, USA), and centrifuged at 100,000× *g* for 2 h at 4 °C on a Beckman Optima L-70 ultracentrifuge (Beckman Coulter GmbH, Krefeld, Germany) using a 40 Ti Swinging-Bucket Rotor. Resulting 100,000× *g* pellets were resuspended in 400 µL of 0.22-µm-filtered PBS and split in half for direct method comparison described below.

### 4.6. SEV Isolation on Sucrose Density Cushion

This protocol was adapted from a previous protocol established in our lab [19]. The half volume of the resuspended 100,000× *g* pellet was filled up with 0.22-µm-filtered PBS to 7 mL. The diluted pellet fraction was carefully applied onto 4 mL of a 40% sucrose cushion with a density of 1.12 g/mL without disturbing the cushion and centrifuged at 100,000× *g* for 2 h at 4 °C. The most upper PBS phase of around 6.5 mL was discarded. The following 3.5 mL high-density sucrose fraction containing the sEVs was recovered. The pellet was left untouched to avoid contaminating the sEV fraction with high molecular weight protein complexes. The sEVs were recovered by washing in 0.22-µm-filtered PBS by adding 7 mL of 0.22-µm-filtered PBS and centrifugation at 100,000× *g* for 2 h at 4 °C. The resulting sEV pellet was resuspended in 250 µL of 0.22-µm-filtered PBS.

### 4.7. SEV Isolation on Single qEV 35nm Columns, IZON

Single qEV 35 nm columns (Izon, Christchurch, New Zealand) were allowed to reach room temperature for 30 min. The resuspended pellet fraction (200 µL) was added onto the column. As soon as the sample volume was taken up by the column, 0.22-µm-filtered PBS was added to the top of the column tube. The following fractions were collected: F0 (1 mL = void volume of the column) and F1 to F7 (250 µL each), according to the manufacturer’s instructions.

### 4.8. Bicinchonic Acid (BCA) Assay and Nanoparticle Tracking Analyzis (NTA)

Protein concentration of sEV samples was assessed employing Pierce™ BCA Protein Assay Kit (Thermo Fisher Scientific Inc., Waltham, MA, USA). 9 µL of each sEV sample was lysed with 1 µL of 10× RIPA buffer (Abcam, Cambridge, UK) and incubated for 30 min on a rotating wheel at 4 °C. Samples were then centrifuged at 17,000× *g* for 20 min at 4 °C. Resulting supernatants were subjected to the BCA assay according to the manufacturer’s instructions. Absorbance was assessed with the use of a MITHRAS LB 940 plate reader (Berthold Technologies, Bad Wildbad, Germany). Particle quantification of sEV samples was performed via NTA using NanoSight LM10 equipped with a 405 nm laser (Malvern Instruments, Malvern, UK). For the NTA analysis, samples were diluted 1:500 to 1:1000 in 0.22-µm-filtered PBS. Camera level and detection threshold were set up at 13 and 5, respectively. The absence of background was verified using 0.2-µm-filtered PBS. For each sample, four videos of 60 s each were recorded and analyzed using the NTA 3.0 software version (Malvern Instruments, Malvern, UK).

### 4.9. Immunoblotting

SEVs and respective parental cells were lysed in RIPA buffer (Abcam, Cambridge, UK), and whole protein lysates were quantified via BCA™ Protein Assay Kit (Thermo Fisher Scientific Inc., Waltham, MA, USA). Per lane, 2.8 µg (human samples) or 3.4 µg (mouse samples) of protein were loaded onto 10% polyacrylamide gels. Following SDS-PAGE and protein transfer, membranes were blocked in 5% bovine serum albumin in Tris-buffered saline (TBS)-Tween 0.1%, and primary antibodies against FLOTILLIN-1 (1:1,000, Cell Signaling Technology, Danvers, MA, USA, #18634), CD81 (1:400, ProSci Inc., San Diego, CA, USA, #5195), CD9 (1:1000, Cell Signalling Technology, Danvers, MA, USA, #13174), TSG101 (1:1000, BD Bioscience, San Jose, CA, USA, #612697), ALIX (1:1000, Cell Signalling Technology, Danvers, MA, USA, #2171) CALNEXIN (1:500, GeneScript, Piscataway, NJ, USA, #A0124040), CYTOCHROME C (1:750, GeneScript, Piscataway, NJ, USA, #A0150740), GM130 (1:1000, Cell Signaling Technology, Danvers, MA, USA, #12480), ATP5A (1:1,000, Abcam, Cambridge, UK, #ab14748) were used in indicated dilutions in 5% bovine serum albumin in TBS-Tween 0.1%. Signals were visualized after secondary antibody hybridization by chemiluminescence detection reagent (Bio-Rad Lab, Hercules, CA, USA, #1705061) with GE Healthcare Amersham Imager 600 (GE Healthcare, Chicago, IL, USA).

### 4.10. Electron Microscopy (EM)

SEV fractions were adsorbed onto glow discharged carbon coated grids, washed in aqua bidest and negatively stained with 2% aqueous uranyl acetate. For immuno-EM, carbon-coated formvar grids were used and the immune reaction was performed after buffer wash including incubation with blocking agent (Aurion, Wageningen, The Netherlands), dilution series of primary antibody HLA-DR (Santa Cruz, Dallas, TX, USA #sc-51618) and Protein A-Au reporter (CMC, UMC Utrecht, The Netherlands). Micrographs were taken with a Zeiss EM 910 or EM 912 at 100 kV (Carl Zeiss, Oberkochen, Germany) using a slow scan CCD camera (TRS, Moorenweis, Germany).

### 4.11. Functional Assay

Murine monocytes were isolated from the bone marrow of C57BL/6 mice by magnetic depletion (EasySep™ Mouse Monocyte Isolation Kit, STEMCELL Technologies Inc., Vancouver, BC, Canada). 5 × 10^4^ cells were cultured in 48-well plates in sEV-free RPMI medium and treated for 8 h with 5 µg of the respective sEV fractions referred above, as determined by BCA assay. Changes in PD-L1, HLA-DR and ICAM-1 expression were evaluated by flow cytometry (BD LSR Fortessa, BD Biosciences, San Jose, CA, USA). The following antibodies were used: PD-L1-PerCP (Biolegend, San Diego, CA, USA, #46-5982-82), HLA-DR-AlexaFluor700 (eBiosciences, San Diego, CA, USA, #56-5321-82), CD54-PE (Biolegend, San Diego, CA, USA, #116108), CX3CR1-BV711 (Biolegend, San Diego, CA, USA, #149031), Ly6C-APC-Cy7 (Biolegend, San Diego, CA, USA, #128015), CD11b-PeCy7 (Biolegend, San Diego, CA, USA, #101216), F4/80-FITC (Biolegend, San Diego, CA, USA, #123107), and the viability dye eFluorTM 506 (eBiosciences, San Diego, CA, USA, #65-0866).

### 4.12. Statistical Analysis

Results of the functional analysis were analyzed for statistical significance with GraphPad PRISM 8.0 software (GraphPad Software, San Diego, CA, USA), using one-way analysis of variance (ANOVA), followed by Tukey’s multiple comparisons. The differences between means were considered significant if *p* ≤ 0.05. The results are expressed as the means ± standard deviation.

### 4.13. EV Track

We have submitted all relevant data of our experiments to the EV-TRACK knowledgebase (EV-TRACK ID: EV200073) (Van Deun J, et al. EV-TRACK: transparent reporting and centralizing knowledge in extracellular vesicle research. Nature methods. 2017;14(3):228–32).

You may access and check the submission of experimental parameters to the EV-TRACK knowledgebase via the following URL: http://evtrack.org/review.php. Please use the EV-TRACK ID (EV200073) and the last name of the first author (Bordas) to access our submission.

## Figures and Tables

**Figure 1 ijms-21-05586-f001:**
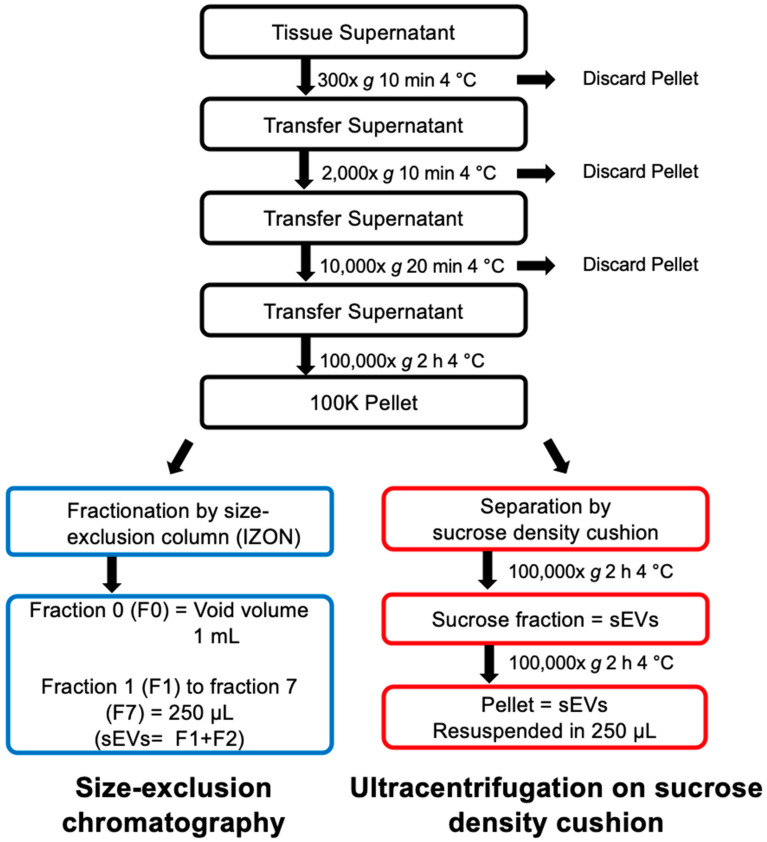
Experimental outline of the comparison of size-exclusion column-based (SEC) versus density-based small extracellular vesicle (sEV) isolations. Supernatant of dissociated lymphatic tissues was separated by differential ultracentrifugation and the resulting 100 K pellet was resuspended and split into two equal parts for direct method comparison. Equal volumes were loaded on either SEC columns or on a sucrose density cushion. Resulting sEV fractions were compared for yield, purity and functionality.

**Figure 2 ijms-21-05586-f002:**
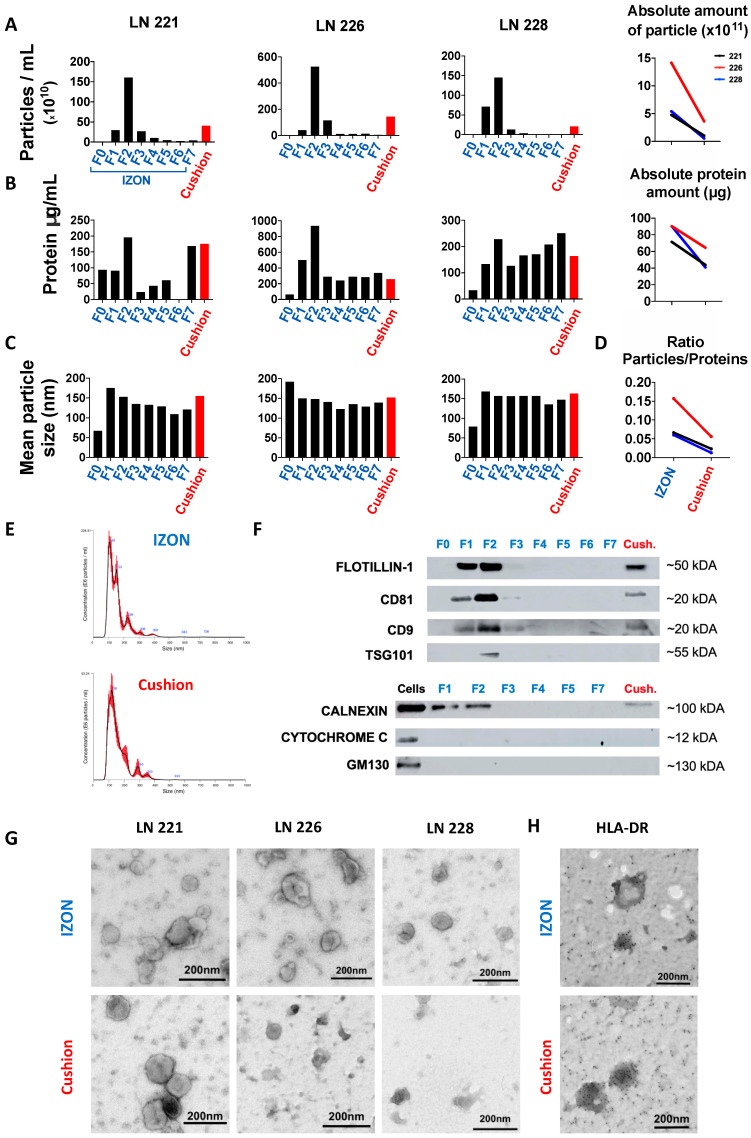
Comparative isolation and characterization of sEVs from human lymph nodes (LN). Size-exclusion column (SEC) fractions (F0 = void volume, 1 mL; F1, F2, F3, F4, F5, F6, F7 = serial fractions, 250 µL) and cushion fraction (pellet resuspended in 250 µL) were analyzed by nanosight tracking analysis (NTA), bicinchoninic acid (BCA) protein quantification, immunoblotting, and transmission electron microscopy (TEM). (**A**) Left: particle concentrations in IZON fractions F0–F7 and “cushion” fraction for three different human LN samples measured by NTA. Right: Absolute number of detected particles as sum of fraction 1 and fraction 2. For each sample, the particle concentration was normalized to the final volume of elution. (**B**) Left: BCA protein quantification for IZON fraction F0–F7 and the “cushion” fraction. Right: Absolute amount of protein in fraction 1 and fraction 2 (the protein concentration was normalized to the final volume of elution). (**C**) Mean particle size for IZON fraction F0–F7 and the “cushion” fraction analyzed by NTA. (**D**) Ratios of particles per protein amount are plotted for IZON and “cushion” fraction. (**E**) Representative particle distribution profile for IZON fraction 2 sample (left) and “cushion” sample analyzed by NTA (Sample LN221). (**F**) Immunoblotting analysis of FLOTILLIN-1, CD81, CD9, TSG101, CALNEXIN, CYTOCHROME C, and GM130 for indicated IZON fractions, the “cushion” fraction and parental cell lysates for one LN sample. (**G**) TEM images of IZON fraction 2 and the “cushion” fraction for the three indicated samples. (**H**) Immunogold electron microscopy for HLA-DR of one human LN sample (sample LN221). Scale bar: 200 µm.

**Figure 3 ijms-21-05586-f003:**
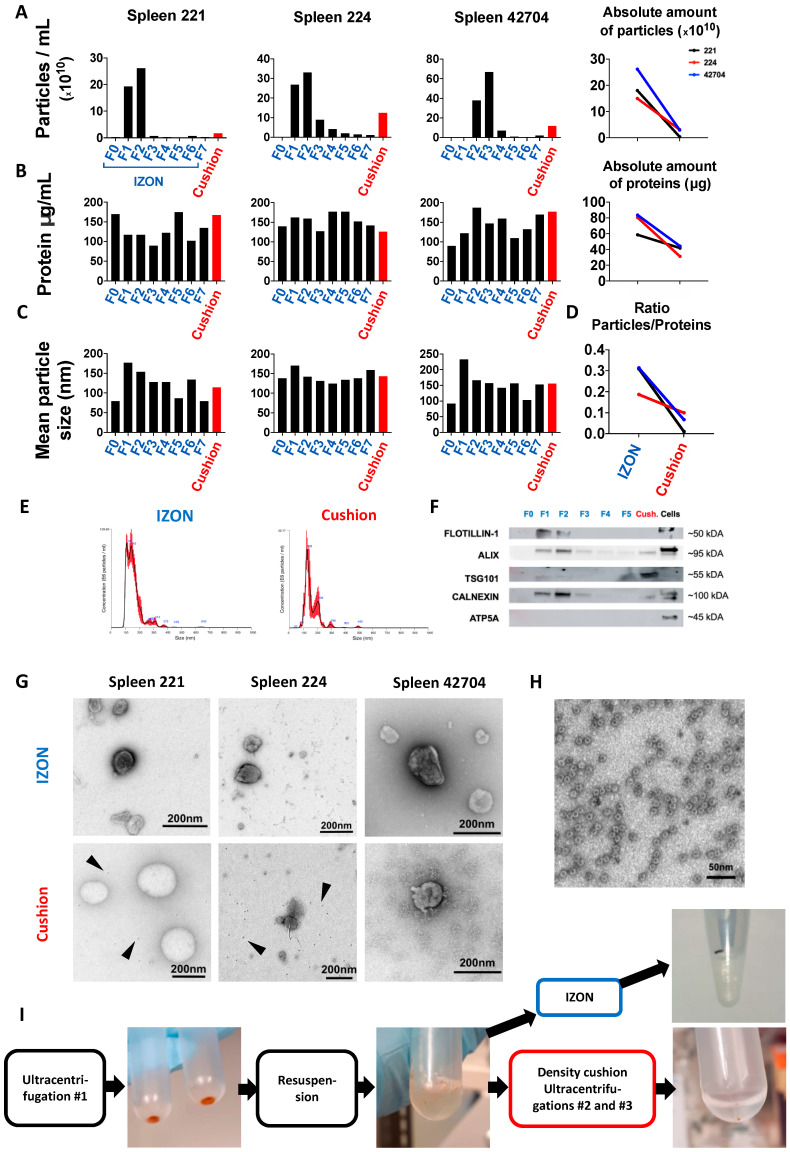
Isolation and characterization of murine spleen sEVs. (**A**) Left: SEV concentration in the different IZON Fractions and “cushion” preparations for three samples analyzed by NTA. Right: Absolute number of particles in indicated preparations. For each sample, the particle concentration in the two peak fractions or in the cushion product was normalized to the final volume of elution. (**B**) Left: protein quantification in the indicated preparations assessed by BCA assay. Right: absolute amount of protein in indicated preparations (the protein concentration was normalized to the final volume of elution). (**C**) Mean particle size of all fractions and the “cushion” preparations analyzed by NTA. (**D**) Ratios of particles per protein amount are plotted for IZON and “cushion” fraction. (**E**) One representative particle distribution profile for an IZON fraction 2 (left) and a “cushion” preparation analyzed by NTA (Spleen 42704). (**F**) Immunoblotting analysis of FLOTILLIN-1, ALIX, TSG101, CALNEXIN, and ATP5A for the different IZON fractions, the “cushion” preparation and parental cells for one spleen sample (spleen 224). (**G**) Transmission electron microscopy (TEM) images of IZON peak fraction and “cushion” preparation for the three indicated samples. (**H**) TEM image of ferritin-like structures found in “cushion” preparations. (**I**) Pictures of the sEV pellet, resuspended sEVs prior to application on the sucrose density cushion, and final pellet in PBS before resuspension.

**Figure 4 ijms-21-05586-f004:**
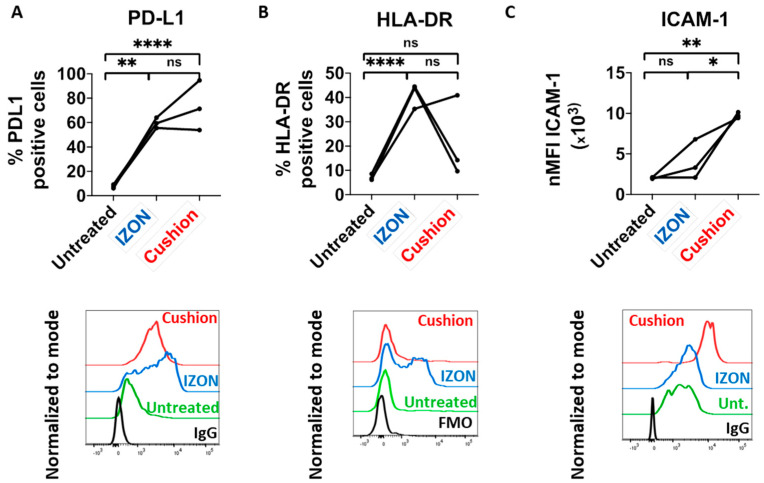
Response of murine monocytes upon tumor-derived sEVs (TEX) treatment. Bone marrow-derived monocytes were treated with 5 µg of the indicated sEV preparations for 8 h and analyzed by flow cytometry gating on CD11b+F4/80+^+^CX3CR1+Ly6C+ cells (*n* = 3 mice per sEV preparation). (**A**) Top: percentage of PD-L1 positive cells among CD11b+F4/80+CX3CR1+Ly6C+ monocytes. Bottom: representative histogram including isotype antibody staining as negative control (IgG). (**B**) Percentage of MHC-II/HLA-DR positive cells among CD11b+F4/80+CX3CR1+Ly6C+ monocytes. Bottom: representative histogram including fluorescence-minus-one (FMO) staining as negative control. (**C**) Top: ICAM-1/CD54 expression presented as normalized mean fluorescence intensity (nMFI). Bottom: representative histogram. *p*-values were determined by one-way ANOVA with Tukey’s multiple comparisons test. * *p* < 0.05; ** *p* < 0.0021; *** *p* < 0.0002; **** *p* < 0.0001.

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
