# Peer review of "Optimized Protocol for Isolation of Small Extracellular Vesicles from Human and Murine Lymphoid Tissues"

_ijms, 2020, doi:10.3390/ijms21155586_

Round 1

Reviewer 1 Report

This is a well written manuscript addressing a fundamental problem in EV biology.  The finding should further unify the field enabling more consistent and reproducible data between different labs movie forward.

Author Response

We would like to thank the referee for reviewing our manuscript and are happy to hear that it is well written and addresses an important topic. 

Reviewer 2 Report

The manuscript is relevant and will be of interest to specialists working in the field of regenerative medicine, transpoantology, reconstructive surgery. I recommend to accept it after minor revision (corrections to minor methodological errors and text editing).

Author Response

We would like to thank the referee for reviewing our manuscript and are happy to hear that the covered topic and presented data are of relevance and interest to a broad range of scientist.

As recommended, we have performed text editing and corrections of methological errors.

Reviewer 3 Report

The article entitled “Optimized protocol for isolation of exosomes from human and murine lymphoid tissues” compare two methods to obtain EVs from solid tumor, from spleen and lymph node. The insights they obtain are useful for researchers and the technology they describe could be applied in the future for other groups.

In my opinion, the correction of some minor details can enhance the strength of the article.  

  1. The most interesting point of the paper is the protocol to extract from the tissue the EVs. However, it Is not clearly described in the M&M. How many amount of lymph nodes are placed in 50 ml of RPMI media? To which size should be minced? How long are incubated? There is antibiotics to prevent bacterial growth? In the case of spleens, how much are necessary to collect enough EVs? Could you briefly describe the MACS disaggregation protocol? I would not mind to see a simple figure with this detail, so readers can approach themselves the purification of EVs from this tissues.
  2. As the authors explain, exosomes are vesicles that come from endocytic pathway. When they isolate vesicles applying the different techniques, they have not evidence that this are exosomes or other types of vesicles. They perform a separation on density or on size, and therefore, in any case they could be certain of the origin of the vesicles. Please rephrase the text by naming EVs, and in the discussion you could argue that they are enrich in the size or the density applied.
  3. Figure 2G, in the legend you can mention if the sample correspond to cushion or SEC. Moreover, I only can see a gold particle in the lower panel.
  4. The claim in lines 242-244 cannot be support by data. “In particular, our results strongly suggest that isolation of exosomes by the sucrose density 242 cushion isolation protocol results in a more severe co-isolation of protein complexes with the 243 exosomes”. The ratio protein/vesicles is similar with both purification methods, and WB for lymph node is quite similar. Certainly, for spleen gradient techniques copurify ferritins, which most likely indicates that density of this structures are similar to EVs.
  5. Regarding differences of effect, authors claim that activation of ICAM can be due to protein contamination. Since they also observe activation of ICAM in one sample obtained by SEC, there is any correlation between activation of ICAM and protein within this group. In another hand, can the authors test their hypothesis about cushion effect by western blot? “These results raise the hypothesis that soluble pro-inflammatory 287 cytokines, secreted in B-cell lymphoma microenvironments, might contribute to the contamination 288 in the “cushion” preparations.”

From my point of view, the discussion should mention that since SEC separates by size, and density gradients separate by density, it is likely that the types of vesicles captured by both techniques could be different. While the characteristics for lymph node EVs are similar for both types of protocols /and it can be claimed better yield for SEC, in the spleen the populations enriched seems to be different. For cushion purified samples, there is a lack of several “EV” markers in the cushion preparations, but an increase of TSG101.

Regarding apolipoprotein, the authors already mention that they did not test the presence of apolipoproteins, and the EM imagen produced is not definitive at this respect. I will be cautious to claim the contamination of cushion samples with these molecules. In theory LDL should copurify with SEC techniques because of their size and HDL will copurify with samples obtained by cushion. It is also likely that the first ultracentrifugation get ride of LDLs, though.

Author Response

We would like to thank the referee for a thorough revision of our manuscript and for helpful suggestions and comments.

Based on these comments, we have revised our manuscript as outlined below.

  1. We would like to thank the referee for pointing out this lack of description. Concerning the processing of the lymph node samples, more details have now been added to the manuscript, including the solution in which the samples were kept, as well as the size of the LN pieces. This additional information ensures that the whole protocol is now fully reproducible. Additionally, we also included more details concerning the processing of spleen samples. As the dissociation is a simple one-step process and always performed according to manufacturer’s instructions, we considered that an additional figure would not bring additional information. 
  2. We completely agree with the referee that based on the provided data, we cannot be sure that we are only analyzing exosomes. As the used protocols do not ensure that other non-exosomal vesicles of small size are co-isolated with exosomes, we exchanged the term “exosome” to “small extracellular vesicles” throughout the text, which is in line with the MISEV guidelines. As shown by our NTA analyses as well as electron microscopy pictures, the size of the isolated particles does not exceed 200 nm, excluding the presence of large extracellular vesicles, like microvesicles.
  3. We corrected the legend of Figure 2G to ensure a proper display of data. Thanks for pointing this out. The poor display of gold particles which should be clearly visible in both the upper and lower picture in this figure is most likely due to low resolution of the figure and hopefully resolved in the final high resolution format of the paper.
  4. We thank the referee for pointing this out. To strengthen this statement, we have now added a supplementary figure to the manuscript displaying the ratio particle/protein, and showing a clear difference between IZON preparations and their respective “cushion” preparations for each lymph node and spleen sample.
  5. We hypothesize in the discussion that soluble cytokines might be a major source of contamination for sEV preparations using the density cushion approach, as they are known to induce ICAM-1 expression on monocytes, and considering that the microenvironment in the murine CLL model we use is enriched in proinflammatory cytokines (Dürr et al, Hematologica 2018, Bresin et al. Cell Death Dis 2016, Efanov et al. Leukemia 2015). We cannot exclude that other sources of contamination may also be responsible for the observed upregulation of ICAM-1. Identifying a specific contaminant in our “cushion” preparations would require additional experiments (e.g. multiplex Elisa or mass spectrometry) that cannot be performed in a short period of time and is therefore beyond the scope of this manuscript.

We agree with the referee that both isolation techniques might lead to the isolation of different subpopulations of sEVs, as we briefly mentioned in the initial manuscript. Additional elements have been added to the discussion to underline this point which is important to keep in mind for the furture use of the described protocols.

As we did not perform experiments to identify apolipoproteins that may be present in our preparations, we decided to regroup under apolipoproteins both LDL and HDL as sources of contamination. Further identification of such contaminants will be done and presented as follow-up work of this manuscript.